# The IFSD Score—A Practical Prognostic Model for Invasive Fungal Spondylodiscitis

**DOI:** 10.3390/jof10010061

**Published:** 2024-01-12

**Authors:** Chao-Chun Yang, Ming-Hsueh Lee, Chia-Yen Liu, Meng-Hung Lin, Yao-Hsu Yang, Kuo-Tai Chen, Tsung-Yu Huang

**Affiliations:** 1Department of Neurosurgery, Chang Gung Memorial Hospital, Chiayi 61363, Taiwan; fergieholic@gmail.com (C.-C.Y.); ma2072@gmail.com (M.-H.L.); tad91116@gmail.com (K.-T.C.); 2Health Information and Epidemiology Laboratory, Chang Gung Memorial Hospital, Chiayi 61363, Taiwan; qchiayen@cgmh.org.tw (C.-Y.L.); mattlin@cgmh.org.tw (M.-H.L.); r95841012@cgmh.org.tw (Y.-H.Y.); 3School of Chinese Medicine, College of Medicine, Chang Gung University, Taoyuan 33302, Taiwan; 4Department of Traditional Chinese Medicine, Chang Gung Memorial Hospital, Chiayi 61363, Taiwan; 5Division of Infectious Diseases, Department of Internal Medicine, Chang Gung Memorial Hospital, Chiayi 61363, Taiwan; 6College of Medicine, Chang Gung University, Taoyuan 33303, Taiwan; 7Microbiology Research and Treatment Center, Chang Gung Memorial Hospital, Chiayi 61363, Taiwan

**Keywords:** invasive fungal infection, fungus, Candida, Aspergillus, spondylodiscitis, osteomyelitis, mortality, Chang Gung Research Database

## Abstract

**Objectives:** Invasive fungal spondylodiscitis (IFSD) is rare and could be lethal in certain circumstances. The previous literature revealed limited data concerning its outcomes. This study aimed to establish a risk-scoring system to predict the one-year mortality rate of this disease. **Methods:** A total of 53 patients from a multi-centered database in Taiwan were included in this study. All the clinicopathological and laboratory data were retrospectively analyzed. Variables strongly related to one-year mortality were identified using a multivariate Cox proportional hazards model. A receiver operating characteristic (ROC) curve was used to express the performance of our IFSD scoring model. **Results:** Five strong predictors were included in the IFSD score: predisposing immunocompromised state, the initial presentation of either radiculopathy or myelopathy, initial laboratory findings of WBC > 12.0 or <0.4 10^3^/µL, hemoglobin < 8 g/dL, and evidence of candidemia. One-year mortality rates for patients with IFSD scores of 0, 1, 2, 3, and 4 were 0%, 16.7%, 56.3%, 72.7%, and 100%, respectively. The area under the curve of the ROC curve was 0.823. **Conclusions:** We developed a practical scoring model with easily obtained demographic, clinical, and laboratory parameters to predict the probability of one-year mortality in patients with IFSD. However, more large-scale and international validations would be necessary before this scoring model is commonly used.

## 1. Introduction

Spinal infections comprise approximately 2.2/100,000 cases per year [1,2,3]. Out of these, infections involving the intervertebral disc and adjacent vertebral bony structures are classified as spondylodiscitis, which accounts for 2–7% of all cases of spinal infections [4,5]. The pathogen of spondylodiscitis can be pyogenic, granulomatous (tuberculosis, brucellosis, fungal infection), or parasitic [6,7]. The incidence of opportunistic fungal spondylodiscitis has surged during the recent four decades because the number of patients with immunocompromised status has grown statistically (approximately 1% of all infectious spondylodiscitis cases [4,8]. The most common species are *Candida*, *Aspergillus*, *Cryptococcus*, and *Coccidioides* [1,9,10,11,12,13,14]. The timeous diagnosis of fungal spinal infections is often hindered by its discrete nature and non-specific symptoms. If the treatment is delayed, it could lead to irreversible neurologic deficits, deformities, systemic infection, and even mortality [15,16]. The overall mortality rate is around 20% [15,17]. Several risk factors for developing fungal spondylodiscitis were investigated, mainly related to immunocompromised states [16,18].

In 2002, clinical experts formed a consensus committee named the European Organization for Research and Treatment of Cancer/Invasive Fungal Infections Cooperative Group and the National Institute of Allergy and Infectious Diseases Mycoses Study Group. They had issued a standard definition for invasive fungal infections [19]. The probability of the diagnosis of invasive fungal infection was classified into three tiers: “proven”, “probable”, and “possible”. “Proven” IFSD requires evidence of fungus species by culture of tissue or pus taken from a disease site without concurrent infection of other organisms. By contrast, “probable” IFSD indicates an immunocompromised host with ongoing fungal sepsis originating from the respiratory tract, the urinary tract, the gastrointestinal tract, and the central nervous system, whereas “possible” IFSD is preferred classification based on appropriate host factors and clinical evidence but without mycological support. Despite several revisions that have been made over the recent 20 years, these updated definitions have yet to be proven applicable in clinical research of a broader range of patients at high risk of immunocompromised states [20,21,22,23,24]. Other predictors had also been evaluated for their association with mortality, including age, sex, clinical findings of neurological deficits, laboratory data, and different treatment strategies. However, the results could have been more satisfying with statistical significance [25]. As a result, we focused only on the “proven” IFSD. We performed a multicentered retrospective study to construct a risk-scoring system based on easily obtained demographic, clinical, and laboratory parameters to predict the probability of one-year mortality in patients with IFSD.

## 2. Materials and Methods

### 2.1. Chang Gung Research Database (CGRD)

We collected patient data from Chang Gung Memorial Hospital (CGMH), which has more than 10,000 beds and admits more than 280,000 patients each year from 2 medical centers, 2 regional hospitals, and 3 district hospitals from the northeast to the south of Taiwan: Keelung, Taipei, Linkou, Taoyuan, Yunlin, Chiayi, and Kaohsiung. The Chang Gung Research Database (CGRD) harbors medical records from those seven medical institutions mentioned above. The basic architecture for the CGRD includes de-identified information from electronic medical records for routine epidemiologic health care studies, with several profiles for laboratory data, in-patient data, out-patient data, emergency room data, pathological data, radiological data, nursing data, charging data, disease category data, surgery data, and mortality outcomes. All the healthcare providers from CGMH can access the data for their clinical practice. Further details about the CGRD were reported [26,27].

### 2.2. Patient Selection

From January 2000 to December 2019, we obtained the data from CGRD of patients with infective spondylodiscitis based on the International Classification of Diseases, ICD-9 codes (722.9, 720.89, 720.9), and ICD-10 codes (M4630-M4659, M4680-M4699). Patients under 18 without antifungal agent use or with antifungal agent treatment for less than one week and without evidence of blood culture, CT-guide biopsy/surgically acquired pus, or tissue culture positive for fungal infection were excluded.

### 2.3. Variables and Outcomes

We divided the patients into two groups based on the primary outcome: one-year mortality. Data including age, sex, predisposing immunocompromised state, initial clinical manifestations and radiological findings, pre-treatment laboratory findings, microbiologic results (blood culture, tissue/pus culture obtained from CT-guide biopsy or open surgical biopsy), different treatment strategies (non-surgical or surgical management), surgically related complications, and in-hospital general complications (respiratory failure, hospital-acquired pneumonia, acute kidney injury, urinary tract infection, electrolyte imbalance, acute coronary syndrome, metabolic encephalopathy, upper gastrointestinal tract bleeding, and pressure sore) were compared between the two groups. The tested variables used for predicting the model of one-year mortality were all recorded at the time of the first diagnosis of IFSD. All table values were displayed as numbers (percentages) or median (interquartile range).

We defined those with predisposing immunocompromised state according to the previous research [12,22,28] as patients who had at least one of the following factors: (1) disturbance of the epithelial barrier caused by broad-spectrum or multiple antibiotic therapies, hemodialysis/peritoneal dialysis, major trauma, and surgery; (2) disease or dysfunction of mononuclear phagocytes or neutrophils, resulting from chemotherapy or radiotherapy, aplastic anemia, chronic granulomatous disease, advanced liver cirrhosis/failure; and (3) defect or dysfunction of T-lymphocyte cell-mediated immunity caused by Acquired Immunodeficiency Syndrome, Hodgkin’s disease, solid organ transplantation, systemic chemotherapy, radiotherapy, hematologic malignancy, and prolonged use of corticosteroids (>0.3 mg/kg for over three weeks).

The clinical neurologic deficits (radicular pain, muscular weakness, myelopathy) and fever were recorded as soon as the symptoms started, whether at the outpatient department, emergency room, or after hospital admission. The onset time for a radiological diagnosis was calculated by weeks. There are five different modalities for diagnosing spondylodiscitis: X-rays, Computed Tomography (CT), Magnetic Resonance Imaging (MRI), Tc-99m methylene diphosphonate bone scan, and Gallium Inflammation Scan. The detailed information included disc space narrowing, endplate erosion, vertebral body collapse, translation or distraction for X-rays; endplate erosion, vertebral body collapse, paraspinal tissue abscess, epidural abscess for CTs; loss of intradiscal key sign, T2 edema signal change, cord/sac compression, root compression, and paraspinal tissue abscess for MRIs. Surgical interventions for IFSD include laminectomy/discectomy (decompression), transpedicular screw insertion (fixation), and posterolateral/transforaminal interbody fusion (fusion), depending on the severity and stage of this disease. In some instances, patients received two or more surgeries during a single hospitalization.

### 2.4. Statistical Analysis

We used Pearson’s chi-squared test to analyze categorical variables and the Wilcoxon rank-sum test to analyze continuous variables. To avoid bias among this mixture of straight and binary variables, the multivariate Cox proportional hazards model was used to analyze the effect of those selected time-to-event predictors on one-year mortality. The outcome was also graphically displayed according to the Kaplan–Meier method, comparing cumulative events by the log-rank test. Lastly, an ROC curve was used in the graphic to show the performance of our IFSD grading model. The statistical calculations were performed with IBM SPSS Statistics for Windows, Version 28.0, Armonk, NY, USA: IBM Corp. When *p* < 0.05, the differences were considered statistically significant.

## 3. Results

### 3.1. Patient Characteristics and Clinical Manifestations

In total, 53 patients with a definite diagnosis of proven IFSD were retrospectively studied over 20 years. Except for the much higher overall one-year mortality rate (45.3%), the mean age (around 65) and male predominance were compatible with the studies previously reported [16,18,25,29]. Predisposing immunocompromised state was seen in about one-third of all the patients with IFSD involved and possessed one of the striking predictors for one-year mortality (50.0% vs. 17.2%, *p* = 0.011). Fever was the typical initial symptom, whereas neurologic deficits were distinct among patients. About 62.3% of patients experienced radicular pain, followed by muscular weakness (43.4%) and myelopathy (13.2%). The onset time to diagnosis was lengthy, with a median of three weeks. Kimona et al. have mentioned that a delayed diagnosis may hinder biological evidence from cultures and biopsies, reduce IFSD pathogen clearance rate, and negatively affect treatment outcomes [30]. Detailed comparisons of baseline characteristics between two groups stratified by one-year mortality are listed in Table 1.

### 3.2. Radiologic and Laboratory Diagnosis

Five different diagnostic imaging, comprising X-rays (in 44 cases), CTs (in 26 cases), MRIs (in 47 cases), bone scans (in 17 cases), and inflammatory scans (in 22 cases), were used with flexibility. Most infections in this study were in the lumbar spine (37 cases, 69.8%), followed by the thoracic (11 cases, 20.8%), cervical (7 cases, 13.2%), and sacral spine (5 cases, 9.4%) regions. Furthermore, seven of them involved two regions. Miller et al. described a case series of spinal infection by Candida [29], 33 cases (55.9%) affecting the lumbar spine, followed by thoracic (17 cases, 28.8%), combined thoracic to lumbar (6 cases, 10.2%), and cervical (3 cases, 5.1%), which was similar to our findings.

Several initially recorded laboratory indicators, including a higher white blood cell count (12.4 vs. 11.0 × 10^3^/µL, *p* = 0.040), higher C-reactive protein concentration (150 vs. 50 mg/µL, *p* = 0.002), lower hemoglobin (9.0 vs. 10.2 g/µL, *p* = 0.034), lower platelet count (169 vs. 291 × 10^3^/µL, *p* = 0.021), as well as a positive finding of candidemia (66.7% vs. 31.0%, *p* = 0.010), led to one-year mortality with statistical significance. As expected, the majority of infections were due to *Candida albicans*, representing 32 cases (60.4%), followed by *C. tropicalis* (7 cases, 13.2%), *C. parapsilosis* (4 cases, 7.5%), and *C. glabrata* (3 cases, 5.7%). There was only one case (1.9%) for each isolated pathogen: *Aspergillus*, *C. krusei*, *C. lusitaniae*, *Cryptococcus neoformans*, *Debaryomyces hansenii*, and *Fonsecaea pedrosoi*. *Candida albicans* infections were identified more often in the survivors’ group (72.4% vs. 45.8%, *p* = 0.049).

### 3.3. Treatment Strategies and Complications

For the use of antifungal agents in our study, 43 patients (81.1%) were treated with intravenous Fluconazole, followed by 13 patients (24.5%) with Micafungin, 11 patients (20.7%) with Anidulafungin, 10 patients (18.9%) with Amphotericin B, and 2 patients (3.8%) each for the rest: Itraconazole, Voriconazole, Posaconazole, and Caspofungin. The median treatment course for antifungal agents was 27 days (11.0–61.0). The results were similar to the 28-patient case series mentioned before [25]. Thirty-five patients (66.0%) were treated with one or more surgeries for IFSD. The indications for surgical intervention were the need to (1) collect pus or infected tissue for definite diagnosis of IFSD, (2) debride and decompress the involved spinal and neural structures, (3) treat cases refractory to medical treatment with disease progression, and (4) stabilize the unstable spine resulting from the vertebral body, disc disruption, and longitudinal ligaments disruption. Regarding the treatment strategies, the chances of receiving necessary surgical interventions were higher in the survivors’ group (82.8% vs. 45.8%, *p* = 0.005). Eleven patients (31.4%) had a laminectomy solely for decompression. The remaining 24 (68.6%) patients had a combined decompression, fixation, and fusion with instrumentation at the time of their first surgery. The selection of surgical strategies was similar between the non-survivors’ and survivors’ groups (36.4% vs. 37.3% compared to debridement only, *p* = 0.011). Ten patients (28.6%) received only one neurosurgical operation, whereas twenty-five patients (71.4%) received at least two operations, with the chances significantly higher in the non-survivors’ group (80.0% vs. 64.0%, *p* = 0.002).

Two surgical-related complications were recorded. Five cases (9.4%) had superficial wound infection, and seven (13.2%) had uncontrolled infection requiring further debridement. Neither event reached statistical differences between the non-survivors’ and survivors’ groups. As for general complications, there were 19 cases (35.8%) with respiratory failure, 17 cases (32.1%) with hospital-acquired pneumonia, 13 cases (24.5%) with acute kidney injury, 13 cases (24.5%) with urinary tract infection, 18 cases (34.0%) with electrolyte imbalance, 3 cases (5.7%) with acute coronary syndrome, 10 cases (18.9%) with metabolic encephalopathy, 10 cases (18.9%) with UGI bleeding, and 4 cases (7.5%) with pressure sores. All the significant complications were higher in the non-survivors’ group. Five of the above reached statistical difference, including respiratory failure (62.5% vs. 13.8%, *p* < 0.001), hospital-acquired pneumonia (58.3% vs. 10.3%, *p* < 0.001), acute kidney injury (45.8% vs. 6.9%, *p* = 0.001), electrolyte imbalance (50.0% vs. 20.7%, *p* = 0.025), and metabolic encephalopathy (37.5% vs. 3.4%, *p* = 0.002).

## 4. Discussion

Our retrospective study used a strict and robust diagnostic definition combining clinical, radiological, and compulsory mycological identification to prevent selection bias. The following are some significant findings that need to be addressed.

### 4.1. The Previous Studies

Literature searches of the online databases, including PubMed, Web of Science, Cochrane Central Register of Controlled Trials, and EMBASE, were performed. To date, 32 case reports, 5 case series, 1 retrospective cohort, and 2 reviews related to IFSD were reported. Fungi are typically considered harmless organisms and are part of the normal human flora. However, in the presence of impaired host immunity or repeated intravascular access, they can become invasive. IFSD was associated with high morbidity and mortality [31]. IFSD can occur either through hematogenous seeding from a distant infection focus or via a direct extension of a contiguous infection from adjacent organs [32]. Hematogenous dissemination is the primary disease mechanism that brings pathogens to the surrounding vascular network around vertebral bodies and intervertebral discs in adults [33].

In traditional radiology, X-rays of the relevant spinal segment are the first-line tool for patients with the above-mentioned clinical symptoms. The sensitivity and specificity are relatively low at 82% and 57% [34]. However, the disc space narrowing, endplate erosions, and vertebral body collapse may appear days or weeks after the infection starts, depending on the pathogen’s virulence and the disease’s natural course [35,36]. A negative native X-ray does not exclude spondylodiscitis. For patients with contraindications to MRI (devices like a pacemaker and ventriculoperitoneal shunt) or with claustrophobia, Computed Tomography (CT) is the best alternative. The enhanced CT studies provide information on endplate erosion, vertebral body collapse, and paraspinal/epidural abscess. CT is also practical for invasive procedures for IFSD, such as fine-needle biopsy and abscess drain placement [37,38,39]. However, plain radiographs/X-rays typically show erosive and destructive vertebral changes with intervertebral disc space narrowing, but these findings may not be radiographically visible for weeks to months [40]. Although CT scanning indicates instant bony changes and reveals the presence of paravertebral involvement or spinal canal compression [41], the provided information is still insufficient for the following treatment modalities.

MRI with contrast is the gold standard in imaging studies to detect spondylodiscitis. Specificity and sensitivity are marked high at 96% and 92% [42,43]. MRI uses different phases to reveal signal changes over the intervertebral disc, vertebral body, paraspinal tissue, and compression to the neural structures (cord, dural sac, nerve roots) [44]. According to previous studies, IFSD usually resulted in disc narrowing, destruction of the endplates, and inflammation of the paraspinal tissues [45]. These imaging findings are consistent with what we found in our case. Bone scintigraphy with 99 technetium-labeled leukocytes and an inflammatory scan with 67 gallium citrate have low specificity but relatively high sensitivity (up to 86%) for diagnosing spondylodiscitis [46]. The advantage of these two radioactively labeled techniques is the potential to detect other sources of infection.

A definitive microbiological diagnosis of IFSD requires the culture of a biopsy specimen to distinguish pathogens of fungi from bacteria, mycobacteria, and malignancies such as multiple myeloma and metastatic disease. Needle biopsy under fluoroscopy or CT guidance for specimens should target the involved vertebral bodies, intervening discs, or paraspinal soft tissues [47]. If the results are negative, repeated needle or open biopsies are still necessary because empirical therapies for IFSD are vastly different based on the pathogens.

Treatment was commonly delayed because of the difficulty in definite diagnosis, which has been mentioned in other series with an average of 6 weeks (median, 1–11 weeks) [16,48,49,50,51]. Other than *Candida albicans*, the other fungal organisms are discrete with their slow-growing nature, and are sometimes complex to cultivate in cultures. Back pain is the most common clinical manifestation, followed by fever and neurological deficits [52]. Longer delays were correlated with a less favorable neurological outcome [53]. Moreover, several reported factors also affected the outcomes, including the pre-existing immunodeficiency secondary to human immunodeficiency virus infection, the use of immunosuppressive drugs such as glucocorticoids or chemotherapy, the prolonged use of intravenous nutrition support, hemodialysis, recent surgery, burns with disrupted skin barrier, and the presence of neutropenia [45,54,55]. The pathogens isolated through cultures have mostly been Candida, Aspergillus, and Cryptococcus [1,4,13]. All results matched our findings.

Surgical treatment is not mandatory in every IFSD case. No consensus has been reached on the best surgical timing [39,56,57,58]. It should be performed in cases with progressive neurological deficits or with evidence of spinal instability. Even if the infection is uncontrolled, the subsequent surgical treatment should be considered carefully because a complete and radical debridement of all the involved tissue is complex and could exhaust patients and surgeons, and generate unnecessary surgical-related complications. The reoperation may also spread the infection toward the adjacent neural structures. In one study, segment instrumentation and bone grafting are not suggested because they lead to overpressure on adjacent vertebral structures, resulting in further instability and higher chances of biofilm formation [59]. As a result, a sufficient and comprehensive antifungal treatment should be the top priority for patients with IFSD. The optimized management of IFSD remains unclear. The suggested first-line antifungal drugs are Amphotericin B or Fluconazole, with an extended treatment course of at least 6 to 10 weeks [60]. Both agents are equally effective. The total length of antifungal therapy has to be determined. Miller et al. recommend considering treatment completed when the patient matches all three indicators: normalized ESR and CRP, clinically resolved symptoms, and improved inflammatory changes over imaging studies [29]. These recommendations align with the Infectious Diseases Society of America’s practice guidelines for treating candidiasis [60].

Concerning the survival analysis, only 1 cohort study investigates the relationship between the risk factors and the one-year mortality of 28 *Candida* vertebral osteomyelitis patients [25]. This survival analysis revealed that the patients with advanced age (median age, 78 vs. 50 years, *p* = 0.02) and a higher Charlson comorbidity index score (median, 6 vs. 2.5, *p* = 0.001) had a higher one-year mortality rate. There was no difference between the survivors and the non-survivors regarding clinical, radiological, biological, or microbiological findings. No scoring system was proposed previously for the overall survival of patients with IFSD. A clear understanding of the various prognostic factors could provide the patients and their families with a realistic expectation of survival. It could also elevate physicians’ awareness to treat individuals with a higher sum of the score cautiously and aggressively.

### 4.2. Our IFSD Score

We selected the contributive indicators and dichotomized the continuous variables into binary variables for the multivariate Cox proportional hazards analysis (Table 2). Regarding the cut points, simply selecting the medians without investigating each relationship with the outcome might lead to significant bias. As a result, decisions were made as the following: defining age > 65 years old as advanced age, body temperature > 38.3 °C and <36.0 °C as hyperthermia and hypothermia, WBC > 12.0 or <0.4 × 10^3^/µL and platelet < 100 × 10^3^/µL as sepsis based on the International Guidelines for Management of Severe Sepsis and Septic Shock [61], CRP > 100 mg/dL as severe systemic infection based on previous research [62], and hemoglobin < 8 mg/dL as severe anemia based on the World Health Organization’s guidelines [63] for all the continuous variables. The one-year mortality of patients with IFSD was statistically significant with the pre-existing immunocompromised state (HR 3.01, *p* = 0.024), the initial presentation of either radiculopathy or myelopathy (HR 4.04, *p* = 0.012), initial laboratory positive findings of leukocytosis or extreme leukopenia (HR 2.83, *p* = 0.049), severe anemia (HR 4.93, *p* = 0.014), and strongly correlated with the evidence of candidemia (HR 2.78, *p* = 0.052).

Every clinical grading scale balances the simplicity and accuracy of outcome prediction. To make this score clinically applicable, the prognosis-predicting model must be simple enough for every healthcare provider across different specialties, including physicians and nurses from local clinics, district hospitals, and medical centers. As a result, we developed a one-year mortality risk stratification scale (the IFSD score, in Table 3) using the data of 53 IFSD cases with the five top-ranked predictors mentioned above. All five predictors were assigned with one point each. The range of IFSD scores was 0 to 5. One-year mortality rates for patients with IFSD scores of 0, 1, 2, 3, and 4 were 0%, 16.7%, 56.3%, 72.7%, and 100%, respectively (in Figure 1). In total, 4 cases (7.5%) scored 0; 18 cases (34.0) scored 1; 16 cases (30.2%) scored 2; 11 cases (20.8%) scored 3; and 4 cases (7.5%) scored 4. No patient in the CGRD cohort had an IFSD score of 5. However, given that no patient with an IFSD score of 4 survived, an IFSD score of 5 would be expected with an extremely high mortality risk. Each increase in the IFSD score was associated with a marked increase in one-year mortality (*p* < 0.001). The mortality rates rose more than three times from an IFSD score of 1 to 2, indicating a decisive difference in clinical outcomes at this turning point. The time-dependent one-year survival curve is depicted in Figure 2 according to the Kaplan–Meier method. The area under the curve (AUC) of the ROC curve was 0.823 (*p* < 0.001) (Figure 3). For a diagnostic model to be meaningful, the AUC ≥ 0.8 is considered good and representative [64]. As a result, our model generates an excellent performance.

### 4.3. Limitations

The main limitation of our work was its retrospective nature, including enrolment biases, patients lost to follow-up, and missing data. Still, some questions needed to be answered. For instance, the study endpoint, the long-term follow-up of functional outcome, and the recurrence rate were not available in the database. The different clinical experiences of spine surgeons from all seven hospitals led to various treatment options and surgical strategies. The selection and priority of the antifungal agents, the indications for surgical intervention, and the perioperative care cannot be standardized. Also, the rareness of the disease might produce low statistical power due to only a small sample size being possible for examination. More large-scale and international validations are necessary before this scoring model can be commonly applied.

## 5. Conclusions

Invasive spondylodiscitis due to fungi was once rare but is now increasing. Most of our patients were treated according to clinical experience and the current guidelines. It is still being determined if they would have had better outcomes with different treatment and surgical strategies. With 53 patients included over 19 years, this work represents the most extensive report of IFSD in the literature to date. We developed the IFSD score with easily obtained demographic, clinical, and laboratory parameters to predict one-year mortality rate. This model could help physicians to identify patients with IFSD with a greater one-year mortality risk of over 50% at a score over two. We plan to use this scoring system in our daily clinical practice to validate whether this model can predict survival accurately. Further prospective trials should investigate long-term functional outcomes and the disease recurrence rate as endpoints.

## Figures and Tables

**Figure 1 jof-10-00061-f001:**
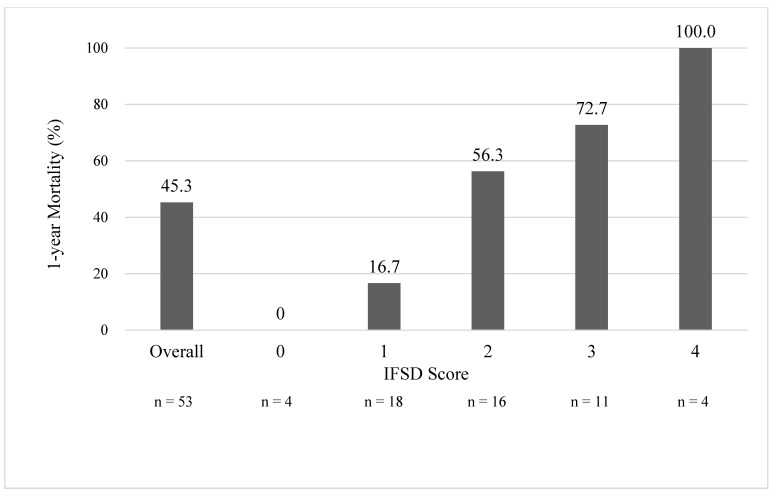
The IFSD score and 1-year mortality. Each increase in the IFSD score was associated with a marked increase in 1-year mortality (*p* < 0.001). No patient in our cohort study had an IFSD score of 5. IFSD, invasive fungal spondylodiscitis. *p* < 0.05, clinical significance.

**Figure 2 jof-10-00061-f002:**
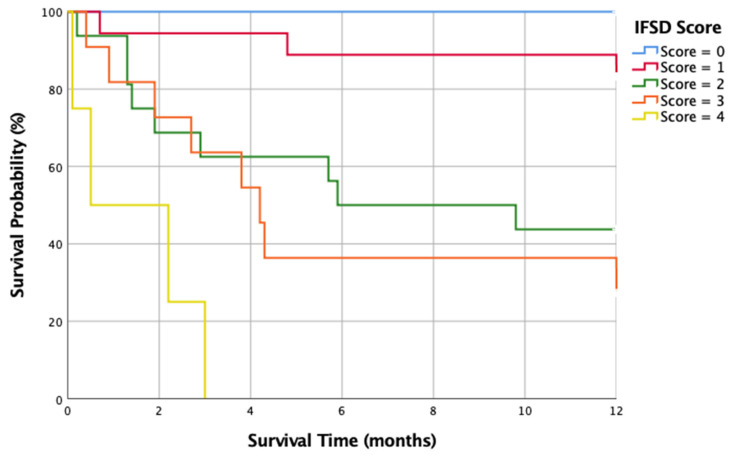
Time-dependent, 1-year-survival Kaplan–Meier curves with different IFSD scores. IFSD, invasive fungal spondylodiscitis.

**Figure 3 jof-10-00061-f003:**
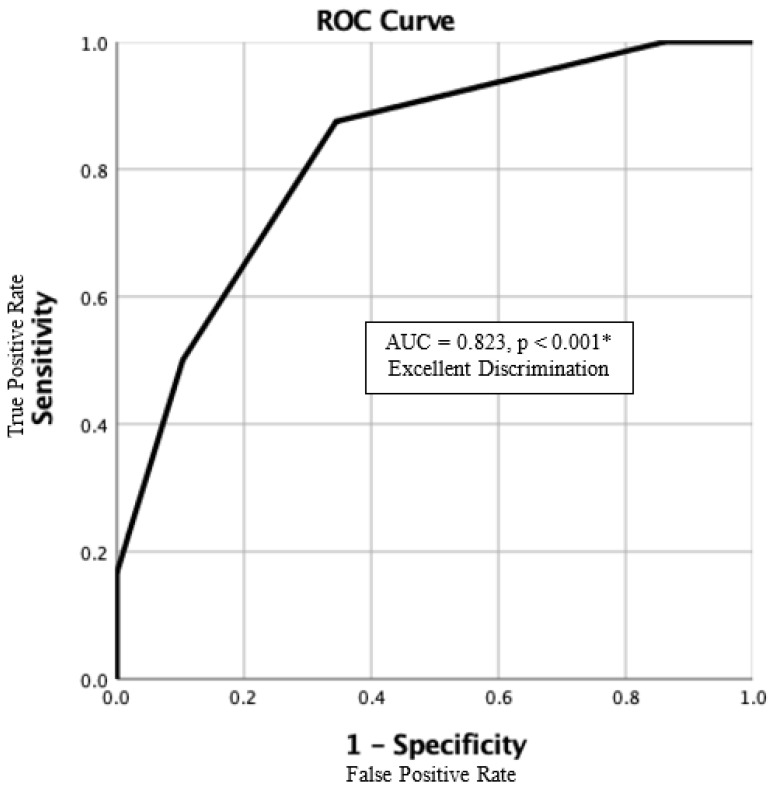
The ROC curve and the AUC. This IFSD scoring model provides excellent performance in predicting a 1-year mortality rate (AUC = 0.823, *p* < 0.001 *). ROC, receiver operating characteristic; AUC, area under the curve; IFSD, invasive fungal spondylodiscitis. * *p* < 0.05, clinical significance.

**Table 1 jof-10-00061-t001:** Clinical and biological characteristics of 53 patients with invasive fungal spondylodiscitis and the survival subgroup analysis. IQ, interquartile; VB, vertebral body; WBC, white blood count; Hb, hemoglobin; ESR, erythrocyte sedimentation rate; CRP, C-reactive protein; UGI, upper gastrointestinal tract. * *p* < 0.05, clinical significance.

	All	One-Year Mortality		
		Dead (*n* = 24)	Survivors (*n* = 29)	*p* Value
Age, years, median, interquartile range	65.6 (56.1–74.0)	67.5 (59.1–74.8)	64.0 (49.4–73.8)	0.090
Male, n, %	35 (66.0)	16 (66.7)	19 (65.5)	0.930
Predisposing Immunocompromised State, n, %	17 (32.1)	12 (50.0)	5 (17.2)	0.011 *
Initial Clinical Manifestations				
Fever, n, %	30 (56.6)	15 (62.5)	15 (51.7)	0.431
Radicular pain, n, %	33 (62.3)	15 (62.5)	18 (62.1)	0.974
Muscular weakness, n, %	23 (43.4)	11 (45.8)	12 (41.4)	0.745
Myelopathy, n, %	7 (13.2)	4 (16.7)	3 (10.3)	0.499
Onset time to diagnosis, weeks, median, interquartile range	3.0 (2.0–5.0)	3.0 (1.8–4.0)	3.0 (2.0–8.0)	0.578
Initial Radiological Findings				
X-rays				
Disc space narrowing, n, %	35 (79.5)	12 (75.0)	23 (82.1)	0.572
Endplate erosion, n, %	29 (65.9)	9 (56.3)	20 (71.4)	0.307
VB collapse, n, %	10 (22.7)	2 (12.5)	8 (28.6)	0.221
CTs				
Endplate erosion, n, %	22 (84.6)	12 (80.0)	10 (91.7)	0.867
VB collapse, n, %	15 (57.7)	9 (60.0)	6 (54.5)	0.462
Paraspinal tissue abscess, n, %	22 (84.6)	11 (73.3)	11 (100.0)	0.356
Epidural abscess, n, %	18 (69.2)	7 (46.7)	11 (100.0)	0.019 *
MRIs				
Loss of intradiscal key sign, n, %	44 (93.6)	18 (94.7)	26 (92.9)	0.796
VB signal change T2 edema, n, %	38 (80.9)	15 (78.9)	23 (82.1)	0.785
Cord/Sac compression, n, %	35 (79.5)	12 (75.0)	23 (82.1)	0.572
Root compression, n, %	29 (65.9)	9 (56.3)	20 (71.4)	0.307
Paraspinal tissue abscess, n, %	10 (22.7)	10 (22.7)	8 (28.6)	0.221
Bone Scans				
Positive for spondylodiscitis, n, %	13 (76.5)	5 (55.6)	8 (100.0)	0.001 *
Inflammatory Scans				
Positive for spondylodiscitis, n, %	19 (86.4)	6 (66.7)	13 (100.0)	0.069
Pre-treatment Lab Findings				
WBC, 10^3^/µL, median, IQ range	11.6 (8.2–13.0)	12.4 (8.9–14.2)	11.0 (7.6–12.5)	0.040 *
Hb, g/dL, median, IQ range	9.7 (8.6–10.9)	9.0 (8.2–10.1)	10.2 (9.4–11.2)	0.034 *
Platelet, 10^3^/µL, median, IQ range	257 (134.0–350.0)	169 (101.5–293.5)	291 (179.0–407.0)	0.021 *
ESR, mm/h, median, IQ range	86 (56.5–103)	95 (58.0–112.3)	82 (57.0–101.0)	0.585
CRP, mg/dL, median, IQ range	94 (41.0–162.8)	150 (91.3–195.5)	50 (25.0–112.0)	0.002 *
Candidemia, n, %	25 (47.2)	16 (66.7)	9 (31.0)	0.010 *
Fungal Species				
Aspergillus unspecified, n, %	1 (1.9)	1 (4.2)	0 (0.0)	-
Candida albicans, n, %	32 (60.4)	11 (45.8)	21 (72.4)	0.049 *
Candida glabrata, n, %	3 (5.7)	2 (8.3)	1 (3.4)	0.444
Candida krusei, n, %	1 (1.9)	0 (0.0)	1 (3.4)	-
Candida parapsilosis, n, %	4 (7.5)	2 (8.3)	2 (6.9)	0.844
Candida tropicalis, n, %	7 (13.2)	4 (16.7)	3 (10.3)	0.499
Canda lusitaniae, n, %	1 (1.9)	1 (4.2)	0 (0.0)	-
Candida unspecified, n, %	1 (1.9)	1 (4.2)	0 (0.0)	-
Cryptococcus neoformans, n, %	1 (1.9)	1 (4.2)	0 (0.0)	-
Debaryomyces hansenii	1 (1.9)	1 (4.2)	0 (0.0)	-
Fonsecaea pedrosoi	1 (1.9)	0 (0.0)	1 (3.4)	-
Treatment Strategies				
Non-surgical treatment, n, %	18 (34.0)	13 (54.2)	5 (17.2)	0.005 *
Surgical intervention, n, %	35 (66.0)	11 (45.8)	24 (82.8)	-
Debridement Only, n, %	11 (31.4)	4 (36.4)	7 (37.5)	0.011 *
Debridement + instrumentation, n, %	25 (71.4)	9 (63.6)	16 (62.5)	-
Multiple-stage OP, n, %	24 (68.6)	8 (80.0)	16 (64.0)	0.002 *
1-stage OP, n, %	11 (31.4)	2 (20.0)	9 (36.0)	-
Surgical Related Complications				
Superficial wounds infection, n, %	5 (9.4)	1 (4.2)	4 (13.8)	0.233
Reoperation for debridement, n, %	7 (13.2)	3 (12.5)	4 (13.8)	0.890
General Complications				
Respiratory failure, n, %	18 (33.9)	15 (62.5)	3 (13.8)	<0.001 *
Hospital-acquired pneumonia, n, %	17 (32.1)	14 (58.3)	3 (10.3)	<0.001 *
Acute kidney injury, n, %	13 (24.5)	11 (45.8)	2 (6.9)	0.001 *
Urinary tract infection, n, %	13 (24.5)	7 (29.2)	6 (20.7)	0.475
Electrolyte imbalance, n, %	18 (34.0)	12 (50.0)	6 (20.7)	0.025 *
Acute coronary syndrome, n, %	3 (5.7)	2 (8.3)	1 (3.4)	0.444
Metabolic encephalopathy, n, %	10 (18.9)	9 (37.5)	1 (3.4)	0.002 *
UGI bleeding, n, %	10 (18.9)	7 (29.2)	3 (10.3)	0.081
Pressure sore, n, %	4 (7.5)	1 (4.2)	3 (10.3)	0.397

**Table 2 jof-10-00061-t002:** Investigating predictors for 1-year mortality using a multivariable Cox proportional hazards model. Four of them were statistically significant, and one of them was strongly related. BT, body temperature; WBC, white blood count; Hb, hemoglobin; CRP, C-reactive protein. * *p* < 0.05, clinical significance.

	Hazard Ratio	95.0% Confidence Interval		*p* Value
		Lower	Upper	
Age > 65	1.53	0.63	3.69	0.346
Immunocompromised	3.01	1.15	7.84	0.024 *
BT > 38.3 or <36	2.44	0.84	7.11	0.103
Radiculopathy or Myelopathy	4.04	1.37	11.93	0.012 *
WBC > 1.2 or <0.4	2.83	1.00	7.95	0.049 *
Hemoglobin < 8	4.93	1.39	17.50	0.014 *
Platelet < 100	0.86	0.24	3.17	0.826
CRP > 100	1.90	0.68	5.29	0.218
Candidemia	2.78	0.99	7.77	0.052

**Table 3 jof-10-00061-t003:** The proposed IFSD score. Each predictor is assigned 1 point. The range of IFSD scores is 0 to 5. IFSD, invasive fungal spondylodiscitis; WBC, white blood count; Hb, hemoglobin.

The IFSD Score	
Component	IFSD Score Points
Immune State	
Immunocompromised	1
Immunocompetent	0
Neurological Deficits	
Radiculopathy or Myelopathy	1
None	0
WBC Count (10^3^/µL)	
>12.0 or <0.4	1
0.4 to 12.0	0
Hemoglobin (g/dL)	
<8	1
≧8	0
Evidence of Candidemia	
Yes	1
None	0
Overall IFSD score	Summation of the points above (0–5)

## Data Availability

Data are contained within the article.

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
