# Peer review of "The IFSD Score—A Practical Prognostic Model for Invasive Fungal Spondylodiscitis"

_jof, 2024, doi:10.3390/jof10010061_

Round 1

Reviewer 1 Report

Comments and Suggestions for Authors

Yang and colleagues present a prognostic model for invasive fungal spondylodiscitis, based on a retrospective look at 53 patients from a database in Taiwan.

Comments:

-          In Table 1, 30 patients have endplate erosion on x-ray in the total column, but only 29 patients are listed in the dead/survivor columns.

-          In Table 1, 23 patients have endplate erosion on CT in the total column, but only 22 patients are listed in the dead/survivor columns.   

-          In Table 1, 23 patients have paraspinal tissue abscesses on CT in the total column, but only 22 patients are listed in the dead/survivor columns.   

-          In Table 1, 17 patients have epidural abscesses on CT in the total column, but more than that, 23 patients are listed in the dead/survivor columns.   

-          In Table 1, 30 patients have root compression on MRI in the total column, but only 29 patients are listed in the dead/survivor columns.   

-          In Table 1, in the pretreatment lab findings area, the WBC for the total column is less than for the dead/survivor columns, but that cannot be correct because the total column would be an average of the dead/survivor columns.   

-          In Table 1, 24 patients have debridement + instrumentation in the total column, but more than that, 25 patients are listed in the dead/survivor columns.   

-          In Table 1, 25 patients have multiple-stage OP in the total column, but only 24 patients are listed in the dead/survivor columns.   

-          In Table 1, 10 patients have 1-stage OP in the total column, but 11 patients are listed in the dead/survivor columns.

-          In Table 1, 19 patients have respiratory failure in the total column, but only 18 patients are listed in the dead/survivor columns.

-          The manuscript does not acknowledge a very important publication in this field:  osteoarticular mycoses. Gamaletsou MN, Rammaert B, Brause B, Bueno MA, Dadwal SS, Henry MW, Katragkou A, Kontoyiannis DP, McCarthy MW, Miller AO, Moriyama B, Pana ZD, Petraitiene R, Petraitis V, Roilides E, Sarkis JP, Simitsopoulou M, Sipsas NV, Taj-Aldeen SJ, Zeller V, Lortholary O, Walsh TJ. International Consortium for Osteoarticular Mycoses. Clin Microbiol Rev. 2022 Dec 21;35(4):e0008619. doi: 10.1128/cmr.00086-19. Epub 2022 Nov 30.

-          Several sections (introduction, discussion) are long and without paragraph breaks. The inclusion of paragraph breaks would be helpful.

Author Response

To Reviewer #1

Reviewer:

  1. In Table 1, 30 patients have endplate erosion on x-ray in the total column, but only 29 patients are listed in the dead/survivor columns.
  2. In Table 1, 23 patients have endplate erosion on CT in the total column, but only 22 patients are listed in the dead/survivor columns.
  3. In Table 1, 23 patients have paraspinal tissue abscesses on CT in the total column, but only 22 patients are listed in the dead/survivor columns.
  4. In Table 1, 17 patients have epidural abscesses on CT in the total column, but more than that, 23 patients are listed in the dead/survivor columns.
  5. In Table 1, 30 patients have root compression on MRI in the total column, but only 29 patients are listed in the dead/survivor columns.
  6. In Table 1, in the pretreatment lab findings area, the WBC for the total column is less than for the dead/survivor columns, but that cannot be correct because the total column would be an average of the dead/survivor columns.
  7. In Table 1, 24 patients have debridement + instrumentation in the total column, but more than that, 25 patients are listed in the dead/survivor columns.
  8. In Table 1, 25 patients have multiple-stage OP in the total column, but only 24 patients are listed in the dead/survivor columns.
  9. In Table 1, 10 patients have 1-stage OP in the total column, but 11 patients are listed in the dead/survivor columns.
  10. In Table 1, 19 patients have respiratory failure in the total column, but only 18 patients are listed in the dead/survivor columns.
  11. The manuscript does not acknowledge a very important publication in this field: osteoarticular mycoses. Gamaletsou MN, Rammaert B, Brause B, Bueno MA, Dadwal SS, Henry MW, Katragkou A, Kontoyiannis DP, McCarthy MW, Miller AO, Moriyama B, Pana ZD, Petraitiene R, Petraitis V, Roilides E, Sarkis JP, Simitsopoulou M, Sipsas NV, Taj-Aldeen SJ, Zeller V, Lortholary O, Walsh TJ. International Consortium for Osteoarticular Mycoses. Clin Microbiol Rev. 2022 Dec 21;35(4):e0008619. doi: 10.1128/cmr.00086-19. Epub 2022 Nov 30.
  12. Several sections (introduction, discussion) are long and without paragraph breaks. The inclusion of paragraph breaks would be helpful

Author:

1 to 10. The exact numbers in the total columns and their percentages are corrected. Thank you for pointing out the mistakes.

  1. This publication is quoted in the introduction and the materials and methods sections.
  2. The paragraph breaks are inserted.

Thank you very much for all the valuable suggestions!

Reviewer 2 Report

Comments and Suggestions for Authors

The research paper authored by Chao-Chun Yang et al. delves into an intriguing topic. The article presents the results of a retrospective cohort study on invasive fungal spondylodiscitis and the IFSD Score. I commend the authors for their dedication and express my appreciation for the chance to evaluate their manuscript. The manuscript is crafted in a good manner. Congratulations to the authors on their findings. 

1. There are a couple of redundant sentences, typos, and grammar errors that need to be corrected. I suggest minor english editing.

2. I also suggest that result section of the manuscript to be reassessed and I suggest not to make discussions in this section. This section in my opinion should include just the results of your study. 

The topic is both original and relevant, addressing a critical subject in the field while effectively filling a specific gap.

It provides a unique perspective by presenting findings based on a single center experience, contributing novel insights that differ from other published materials in the subject area.

Author Response

To Reviewer #2

Reviewer:

The research paper authored by Chao-Chun Yang et al. delves into an intriguing topic. The article presents the results of a retrospective cohort study on invasive fungal spondylodiscitis and the IFSD Score. I commend the authors for their dedication and express my appreciation for the chance to evaluate their manuscript. The manuscript is crafted in a good manner. Congratulations to the authors on their findings.

  1. There are a couple of redundant sentences, typos, and grammar errors that need to be corrected. I suggest minor english editing.
  2. I also suggest that result section of the manuscript to be reassessed and I suggest not to make discussions in this section. This section in my opinion should include just the results of your study.

The topic is both original and relevant, addressing a critical subject in the field while effectively filling a specific gap. It provides a unique perspective by presenting findings based on a single center experience, contributing novel insights that differ from other published materials in the subject area.

Author:

  1. The English editing is completed by a native speaker.
  2. I cut down the result section and move some parts to the discussion section.

Thank you very much for all the valuable suggestions!

Reviewer 3 Report

Comments and Suggestions for Authors

My congratulations to the authors because the paper contributes to knowledge in this field, although the sample size, as the authors recognize, is a limitation. It is a condition with a low frequency but that is increasing given the increasing population of immunosuppressed patients. The research design meets the stated objective and provides a tool that will be very useful in the diagnostic and prognostic approach of these patients. The retrospective nature was necessary to be able to collect this type of patients, so I believe that it is not necessarily a limitation in this scenario but rather a characteristic of the condition that was studied given its clinical presentation. My only suggestion is, again, as the authors recognize. be able to implement a prospective validation cohort to evaluate the operational characteristics of the test in real life.

Author Response

To Reviewer #3

Reviewer:

My congratulations to the authors because the paper contributes to knowledge in this field, although the sample size, as the authors recognize, is a limitation. It is a condition with a low frequency but that is increasing given the increasing population of immunosuppressed patients. The research design meets the stated objective and provides a tool that will be very useful in the diagnostic and prognostic approach of these patients. The retrospective nature was necessary to be able to collect this type of patients, so I believe that it is not necessarily a limitation in this scenario but rather a characteristic of the condition that was studied given its clinical presentation. My only suggestion is, again, as the authors recognize. be able to implement a prospective validation cohort to evaluate the operational characteristics of the test in real life.

Author:

We will work on the prospective validations in the near future.

Thank you very much for all the valuable suggestions!

Reviewer 4 Report

Comments and Suggestions for Authors

The authors present a score to predict the outcome of invasive fungal spondylodiscitis.

This is an interesting idea although the prediction of death is not helpful as far as the treatment is concerned, since a high score does not suggest another therapeutic method.

The paper is written in good English language needing only small corrections.

Abstract could better describe the objectives.

Introduction, materials, and methods are well presented.

Results are also well and clearly presented.

But in the section of discussion, the part of previous studies does not have any place according to my opinion. Too long, only the last part should be kept, since it emphasizes the utility of the present study.

And another point is the fact that references are rather old.

Comments on the Quality of English Language

The paper is written in good English language needing only small corrections.

Author Response

To Reviewer #4

Reviewer:

The authors present a score to predict the outcome of invasive fungal spondylodiscitis.

This is an interesting idea although the prediction of death is not helpful as far as the treatment is concerned, since a high score does not suggest another therapeutic method.

The paper is written in good English language needing only small corrections.

Abstract could better describe the objectives.

Introduction, materials, and methods are well presented.

Results are also well and clearly presented.

But in the section of discussion, the part of previous studies does not have any place according to my opinion. Too long, only the last part should be kept, since it emphasizes the utility of the present study.

And another point is the fact that references are rather old.

Author:

  1. The English editing is polished by a native speaker.
  2. Objectives in the abstract section is revised.
  3. The discussion section is revised.
  4. More than 10 recent references between 2018 to 2023 are added to this article for quotations.

Thank you very much for all the valuable suggestions!

Round 2

Reviewer 1 Report

Comments and Suggestions for Authors

The authors have responded to my comments

Reviewer 2 Report

Comments and Suggestions for Authors

The revised version of the manuscript is greatly improved and in my personal opinion is suitable for publishing.

Reviewer 4 Report

Comments and Suggestions for Authors

I have no further comments